

# Environmental and individual determinants of burrow-site microhabitat selection, occupancy, and fidelity in eastern chipmunks living in a pulsed-resource ecosystem

Camille Gaudreau-Rousseau[1], Patrick Bergeron[2], Denis Réale[3] and Dany Garant[1]

[1] Département de Biologie, Université de Sherbrooke, Sherbrooke, Québec, Canada
[2] Department of Biological Sciences, Bishop's University, Sherbrooke, Québec, Canada
[3] Département des Sciences Biologiques, Université du Québec à Montréal, Montréal, Québec, Canada

## ABSTRACT

**Background:** Habitat selection has major consequences on individual fitness, particularly selection for breeding sites such as nests or burrows. Theory predicts that animals will first use optimal habitats or rearrange their distribution by moving to higher-quality habitats whenever possible, for instance when another resident disperses or dies, or when environmental changes occur. External constraints, such as predation risk or resource abundance, and interindividual differences in age, sex and body condition can lead to variation in animals' perception of habitat quality. Following habitat use by individuals over their lifetime is thus essential to understand the causes of variation in habitat selection within a population.

**Methods:** We used burrow occupancy data collected over eight years to assess burrow-site selection in a population of wild eastern chipmunks (*Tamias striatus*) relying on pulsed resources. We first compared characteristics of burrow microhabitats with those of equivalent unused plots. We then investigated the factors influencing the frequency of burrow occupation over time, and the individual and environmental causes of annual burrow fidelity decisions.

**Results:** Our results indicate that chipmunks select microhabitats with a greater number of woody debris and greater slopes. Microhabitats of burrows with higher occupancy rates had a lower shrub stratum, were less horizontally opened and their occupants' sex-ratio was skewed towards males. Burrow fidelity was higher in non-mast years and positively related to the occupant's age, microhabitat canopy cover and density of large red maples.

**Conclusion:** The quality of a burrow microhabitat appears to be determined in part by characteristics that favour predation avoidance, but consideration of occupancy and fidelity patterns over several years also highlighted the importance of including individual and contextual factors in habitat selection studies.

Corresponding author
Camille Gaudreau-Rousseau,
camille.gaudreau-rousseau@usherbrooke.ca

# INTRODUCTION

Habitat selection is the process whereby individuals preferentially use a set of available habitats at multiple spatial scales (*Hutto, 1985*; *Morris, 2003*), leading to a disproportionate use of some resources over others (*Johnson, 1980*). Selected habitats should allow organisms to maximize their survival and reproduction (*Orians & Wittenberger, 1991*), while accounting for various external constraints (*i.e.*, their realized niche; *Hutchinson, 1957*), such as predation and competition (*Cody, 1981*; *Rosenzweig, 1991*; *Brown, 1999*). Therefore, identifying the factors that influence an organism's spatial distribution helps evaluate potential evolutionary pressures at work on habitat selection.

Habitat components driving selection patterns at a given scale can rarely be extrapolated across other scales (*Hall, Krausman & Morrison, 1997*; *Mayor et al., 2009*). Furthermore, in species-habitat association studies, the most informative scale at which to assess habitat selection depends on the species (*Rettie & Messier, 2000*; *Fisher, Anholt & Volpe, 2011*). While habitat selection at the geographic range level is probably genetically determined in a species or population (*Hutto, 1985*), selection inside a home range or territory (*i.e.*, microhabitat; *Johnson, 1980*) reflects individual choices upon which proximal and ultimate causes can be investigated *via* reproduction, life-history traits, behaviors, energetics and survival (*Wiens, 1972*; *Edelaar et al., 2008*). The adaptive significance of habitat-selection decisions also depends on the time frame involved (*e.g.*, seasonal *vs* daily decisions) relative to an animal's lifespan (*Orians & Wittenberger, 1991*; *Mayor et al., 2009*). For territorial species, selection for dens, nests and burrows, which are used intensively and for long periods, have a crucial influence on fitness (*Hansell, 1993*; *Landry-Cuerrier, 2008*). Specifically, microhabitat features can influence food availability and protection against predators (*e.g.*, *Zollner & Crane, 2003*; *Hayes, Chesh & Ebensperger, 2007*). Variation in microhabitat quality due to spatial and temporal heterogeneity in the distribution and quantity of these features is therefore a key factor to consider in habitat selection studies, especially for species whose individuals' fitness heavily depends on the long-term use of a particular site (*Johnson, 2007*; *Potti et al., 2018*).

Theory predicts that occupants of high-quality territories should display high site fidelity, whereas low-quality territory occupants should aim to relocate (*Schmidt, 2014*; *Acker et al., 2017*). However, theoretical models of habitat distribution commonly fail to consider several limitations on individuals' choice, which often leads to over- or under-estimations of site fidelity rates (*Switzer, 1993*; *Piper, 2011*). First, individuals seldom possess perfect information about site quality because search costs and time constraints prevent them from gathering information on all sites (*Orians & Wittenberger, 1991*; *Stamps, 2001*). Also, habitat quality is rarely fixed in time and is affected by intra- and inter-annual variation in environmental conditions (*Morris, 1988*), such as food availability, predator presence or intraspecific competition (*e.g.*, *Rettie & Messier, 2000*; *Hwang, Larivière & Messier, 2007*). Site fidelity should thus increase with the cost of

changing territories and habitat unpredictability (*Switzer, 1993*; *Gerber et al., 2019*). Second, a habitat may not represent the same quality for every individual, because individuals present phenotypic differences and gradually gain private familiarity advantages with the space they occupy, which increases the costs of dispersing (*Piper, 2011*). For example, sex differences in habitat selection are present in several mammals, because females generally compete for available resources, while distribution of breeding partners limits male reproductive success (*Emlen & Oring, 1977*). Age can also play a role on habitat selection: inexperienced juveniles are often less able to recognize a good habitat or to defend a territory against adults, such that site fidelity often increases with age (*Switzer, 1993*). Individual differences in traits related to dominance, such as body mass or aggressivity, can create further discrepancies in habitat selection and site fidelity (*Acker et al., 2017*). Finally, familiarity advantages associated with long-term incumbency at a territory are regularly overlooked (*Piper, 2011*; *Schmidt, 2014*). The perceived value of a site increases with the knowledge a resident acquires about its abiotic and biotic features (*e.g.*, location of food, topography, vegetation, refuges, and conspecific neighbors) (*Bruinzeel & van de Pol, 2004*; *Brown, Bomberger Brown & Brazeal, 2008*; *Piper, 2011*), therefore favoring site fidelity (*Switzer, 1993*).

The eastern chipmunk (*Tamias striatus*) is a small, solitary, hibernating, diurnal rodent of the Sciuridae family found in North American deciduous forests (*Elliott, 1978*; *Snyder, 1982*). Longevity of chipmunks varies between means of 1.5 and 3 years but can be as long as 8 years (*Snyder, 1982*; *Bergeron et al., 2011*). Up to 90% of their diet consists of tree seeds (*Lacher & Mares, 1996*), but they supplement their diet with herbaceous bulbs, invertebrates, mushrooms and berries (*Wrazen & Svendsen, 1978*). Chipmunks individually occupy a burrow that serves as a hibernaculum, a nursery, a larder hoard and a refuge from weather, predators and conspecifics (*Yahner, 1978a*; *Svendsen & Yahner, 1979*). They are more prone to reuse and expand an existing vacant burrow system rather than excavate an entire one anew (*Panuska & Wade, 1956*; *Elliott, 1978*), but when they do, it first consists of a simple system that is gradually enlarged and complexified (*Thomas, 1974*). They can also change burrow system from one year to the next (*Yahner, 1978a*; *Marmet, Pisanu & Chapuis, 2009*). Individual home ranges may overlap widely, but each chipmunk defends an exclusive ~15 m radius core area around its burrow (*Yahner, 1978a*; *Couchoux et al., 2021*). Hence, the burrow's aboveground immediate surroundings and defended area (hereafter called "microhabitat" and defined as a 10 m-radius plot) are characterized by prolonged and intensive use and should thus represent a major determinant of the occupant's fitness. Microhabitat characteristics are therefore expected to influence burrow-site selection and correlate with occupancy rates and site fidelity decisions.

Habitat selection in small mammals has been extensively studied but is frequently limited to broad associations between general space use and microhabitat characteristics (*e.g.*, *Bowers, 1995*; *Coppeto et al., 2006*; *Lindemann, Harris & Keller, 2015*). Few studies have assessed the relative influence of habitat characteristics, variation in environmental conditions, and among-individual differences on microhabitat selection across multiple years. Here, we use burrow occupancy and individual-based data from 2012 to 2019 in a

population of eastern chipmunks characterized by important inter-annual variations in food availability due to masts in southern Québec, Canada, to investigate environmental and individual determinants of burrow-site selection, occupation and fidelity in this species. Masts are recurring events of large and synchronous production of seeds by one or several tree species at a regional scale, generally occurring every 2 to 5 years (*Cleavitt & Fahey, 2017*). In the region, American beech trees (*Fagus grandifolia*) mast approximately every other year (*Tissier et al., 2020*) and play a major role in most aspects of chipmunk's ecology (reproduction: *Bergeron et al., 2011*; dispersal: *Dubuc Messier et al., 2012*; activity: *LaZerte & Kramer, 2016* and life-history: *Montiglio et al., 2014*). They rely on their cached seeds and bulbs to survive hibernation, support spring reproduction and as long-term reserves in case of mast failure (*Humphries et al., 2002*). Chipmunks systematically breed during the summer preceding a mast, such that juveniles emerge from the maternal burrow in the fall when beech nuts are available, and also reproduce the following spring. Such synchronicity between beech masting and chipmunk reproduction yields a 15-months gap with no juvenile production between a spring reproduction and the next pre-mast summer reproduction (*Bergeron et al., 2011*). This pulsed-resource system drives chipmunks population dynamics and offers a unique opportunity to study the effects of varying environmental conditions on habitat selection decisions.

We first investigated the microhabitat features required for burrow-site establishment in this species. Based on previous studies on microhabitat selection by chipmunks, we predicted that burrows would be located in areas containing features that typically enhance predator avoidance, such as woody debris (*Svendsen & Yahner, 1979*; *Landry-Cuerrier, 2008*), low shrub stratum coverage and a high canopy (*Bowers, 1995*; *Mahan & Yahner, 1996*). We also predicted that burrow microhabitats would contain more dietary items required for persistence (*i.e.*, seed-producing trees and herbaceous plants) compared to non-burrow microhabitats. We then investigated if differences in burrow occupancy rates could point to the existence of a quality gradient among microhabitats, and if, in turn, access to that quality could be mediated by sex- and/or age-specific selective behavior in microhabitat characteristics. We hypothesized that juveniles are less selective than adults due to the necessity of quickly finding a vacant burrow upon dispersal from the maternal burrow, especially when population density is high (*Stamps, 2001*; *Dubuc Messier et al., 2012*). We therefore predicted that they would occupy burrows that have low occupancy rates. We also predicted that females would favour microhabitats containing more food resources compared to males given the need for resources to support reproduction (*Greenwood, 1980*). Lastly, we assessed the individual and environmental determinants of year-to-year burrow-site fidelity. We predicted (1) low site fidelity in juveniles given their inexperience and lower competitiveness compared to older, established adults (*Yahner, 1978a*), (2) that body mass should positively relate to site fidelity, because heavier individuals may be more capable of withholding their burrow when facing competition from conspecifics (*Couchoux et al., 2021*), (3) that burrow fidelity should be higher following mast years, as individuals would therefore benefit from the massive amounts of stored seeds that allows them to sustain activity for months (*Elliott, 1978*; *Humphries et al., 2002*) and finally (4) that local density of neighbours around a burrow may affect the

fidelity of its occupant, either positively, if higher density leads to more pressure to keep possession of an earned burrow, or negatively, if it rather leads to an increase in agonistic interactions, intraspecific competition and dispersal behavior.

# MATERIALS AND METHODS

## Study system

We monitored a population of eastern chipmunks from 2012 to 2019 on three sites, two of which were 6.76 ha and one 3.24 ha, located in southern Québec (45°06′N, 72°25′W), Canada (Fig. S1). Sites were located less than 10 km apart and had similar habitats consisting of a mature deciduous forest dominated by masting American beeches, sugar maples (*Acer saccharum*) and red maples (*A. rubrum*). Upon emergence from hibernation in early May, bulbs of the most abundant herbaceous spring-blooming plants on the sites, namely yellow trout lily (*Erythronium americanum*) and Carolina spring beauty (*Claytonia caroliniana*), constitute the main food source. Spring masting of red maple also contributes to chipmunks' diet, especially during the years of summer reproduction (*Tissier et al., 2020*).

We assessed the presence or absence of a beech mast during autumn using permanent plastic buckets (0.06 $m^2$ opening) installed in spring 2012 under 34 beech trees (site 1: 13 trees, site 2: 13 trees and site 3: eight trees) having a circumference at breast height ≥31 cm. To deter seed-predator access, buckets were placed 1 m away from the trunk and mounted on metal posts at >50 cm above ground (see *Tissier et al., 2020*). Their content was collected and counted in October yearly to confirm low-food or high-food observations. We classified years as mast years when production exceeded 50 seeds/$m^2$ (as in *Paquette et al., 2020*).

## Chipmunk capture and monitoring

Throughout their active season, from the end of April until mid-September, we trapped chipmunks at least 1 day per week on each site, using *Longworth* traps (Longworth Scientific Instruments, Abingdon, UK) placed at 40-m intervals in fixed trapping grids (sites 1 and 2 had 98 trap locations, site 3 had 50 trap locations). Traps were checked every 2h from 8:00 a.m. until dusk. At their first capture, individuals were ear-tagged (National Band and Tag Co., New York, KY, USA) and received colour-coded flags temporarily attached to ear tags for remote visual identification, along with a subcutaneous transponder (Trovan®PIT-tag; EIDAP, Inc., Sherwood Park, AB, Canada). At each capture, individuals were identified and body mass, sex, reproductive status, age (juvenile or adult) and location of capture were recorded. Following *Bergeron et al. (2011)*, an individual weighing <80 g upon emergence and showing no signs of reproduction (developed scrotum or mammae) throughout the entire trapping season was considered a juvenile until the following breeding season. Individuals were considered residents when captured more than five times during a season and over a period longer than 2 weeks. Animals were captured and handled in compliance with the Canadian Council on Animal Care, under the approval of the Université de Sherbrooke Animal Ethics Committee (protocol numbers: DG2011-01, DG2015-02, DG2019-01) and the Ministère de la Forêt,

de la Faune et des Parcs du Québec (#2017-05-01-102-05-S-F, #2018-04-20-103-05-S-F, #2019-04-30-104-05-S-F).

## Burrow identification and microhabitat characteristics

The population's peak activity period occurred in June, during which the proportion of marked captured individuals across all years was above 90%. Most chipmunk-burrow associations were thus determined in June yearly by opportunistically fitting resident chipmunks with radio-transmitting collars (PD-2C model; Holohil Systems Ltd, Carp, Ontario, Canada) and tracking them using telemetry between 8:00 and 9:00 p.m., when they are no longer active aboveground. Burrow location was identified where the radio signal was the strongest, presumably corresponding to the burrow's main chamber. An individual was assigned to a burrow when it was tracked to the same location at least twice during the same week, as in *Dubuc Messier et al. (2012)*, or when it was observed (either opportunistically or through individual-targeted observations sessions) carrying food in the burrow at least three times in the same week. Following burrow identification and upon recapture of its associated chipmunk, collars were removed to be placed on other individuals (mean ± SD of 10.26 ± 11.54 days per individual with collar, 90% collar retrieving success). We assumed burrows stayed occupied by the same individual during the remainder of the active season, which, at least for adults, is a fair assumption considering that most movements between burrows occur either during spring or late summer (*Elliott, 1978*; *Yahner, 1978a*). Our systematic burrow identification approach gives us confidence that most burrows were located on our study sites.

For each burrow, microhabitat was assessed within a 10 m radius circular plot centred on the main chamber. Within this circle, we counted all trees higher than breast height, identified them to species and classified them into three categories based on their diameter at breast height (DBH; small: DBH ≤ 10 cm, medium: 10 cm < DBH ≤ 31 cm, large: DBH > 31 cm). From this information, we calculated small and medium hardwood tree density, density of large beech, red maple and sugar maple trees, and average DBH of large seed-producing trees (as in *Landry-Cuerrier, 2008*). We then visually estimated yellow trout lily and Carolina spring beauty cover percentage, percentage of canopy closure and of cover by woody debris, rocks, and plants and shrubs less than one meter high. Canopy height was scored on a scale from 0 to 4 (0 = low height, 1 = mix of low and medium height, 2 = medium height, 3 = mix of medium and high height, 4 = high height). Horizontal openness was also scored on a scale from 0 to 4, zero being very open (very thin or no understory, easy to walk through) and four very close (dense understory, difficult to walk through). The most representative slope in a 1-m radius surface surrounding the burrow entrance was measured using a clinometer (in degrees). Finally, we counted the number of refuges (8 to 15 cm cavities that can shelter a chipmunk, *e.g.*, in a tree stump or under a rock) available in the plot, and the number of logs >2 m long. These different variables allowed us to estimate food availability (variables related to seed-producing trees and grasses), structure (variables related to ground cover, canopy and slope) and the presence of characteristics likely related to predator avoidance (variables concerning refuges and woody debris).

In 2019, we randomly sampled and paired 10-m radius control plots with known burrows (1:1 ratio) to represent available but unchosen microhabitats. Control locations were determined by selecting a point 20 m away in a random direction from a known burrow, which became the center of the control plot. That point could not be within 20 m of another burrow or control plot, in which case a different random direction was determined. Among all burrows discovered from 2012 to 2018, about 30 were randomly selected on each site ($N = 102$; 35 on site 1, 36 on site 2 and 31 on site 3) to be paired with a control plot and were resampled to ensure the validity of the comparisons. We also ensured they were well scattered across the entire study grids. The same environmental variables as for known burrows were collected on these control plots. We assessed microhabitat measurements' stability through time by comparing values from sampling in 2019 to sampling values when the burrow was first discovered (between 2012 and 2018). Environmental variables changed <5% between sampling periods.

## Statistical analyses

### Microhabitat selection

The environmental variables of the 102 burrow microhabitats and control plots were compared using a conditional logistic regression for matched case-control pairs, with the probability of a microhabitat use (0 or 1) as the response variable and the environmental variables previously described as fixed effects, with no interactions. We also tested the second-order quadratic effect of small hardwood tree density, canopy height, slope and horizontal openness, as these variables could be optimal at an intermediate level rather than extreme values, in relation for instance to predator avoidance. Quadratic effects were not significant and were excluded from subsequent analysis.

### Microhabitat occupancy

We first assessed if the frequency of burrow occupancy differed from randomness, which would indicate selection or avoidance of certain microhabitats, and thus a potential quality gradient among them. If burrows were occupied randomly without any selection and independently of previous occupation, we would expect most of them to have an occupancy rate equal to the mean, with a minority being more or less occupied simply by chance. Moreover, no burrow has a null probability of being occupied because, by design, all burrows had to be occupied at least once to be detected. Thus, we estimated the expected burrow occupancy frequencies from a zero-truncated Poisson distribution. Observed frequencies correspond to the number of burrows occupied for 1 to 7 years, respectively. We then tested whether frequencies of observed occupancy differed from expected ones using a Fisher's exact test (as in *Sergio & Newton, 2003*; *Pagán, Martínez & Calvo, 2009*; *Anderson et al., 2019*).

An occupancy score over time was calculated for every burrow by dividing the number of years a burrow was occupied by the number of years since its first detection (*i.e.*, its availability). Determinants of burrow occupancy were tested using a logistic model, with each burrow's occupation score as the response variable and the environmental variables used in the microhabitat selection analysis as fixed effects, except for slope as this measure

was unavailable for most burrows. To account for possible differences in access to microhabitat quality between sexes and ages, we included the proportion of females and the proportion of adult occupants among all chipmunks that occupied a given burrow during the study period. The model was weighted by the number of years since each burrow was discovered. To further investigate possible intersexual difference in microhabitat selection or use, we ran a binomial model with the sex ratio (females/males) of every burrow's successive adult occupants as the response variable, and all microhabitat variables as fixed effects.

### Microhabitat fidelity

We assessed fidelity on a year-to-year basis, with fidelity determined by the identity of a burrow's occupant on two consecutive years. Fidelity was thus defined as a binomial variable (1 if the burrow was kept by the same occupant the following year and 0 if it was retrieved by a different individual or abandoned) and was analysed using a logistic regression. When scored 0, we restricted the analysis to instances where the first occupant was found at another burrow the following year, to avoid biases due to detection and mortality. Four cases of individuals recaptured after more than one year were also included. We tested for the effect of the occupant's characteristics on burrow fidelity by including sex, age (in years, 0 being the juvenile state), birth cohort (summer or spring) and average body mass throughout the trapping season. We included beech seed production (two-level factor: mast or non-mast) and the number of adjacent occupied home ranges overlapping a focal occupant's home range, considering a mean home range size of 40 m radius around a burrow on our study sites (as in *Dubuc Messier et al., 2012*, see also *Mares, Watson & Lacher, 1976*). The number of adjacent home ranges is used as a proxy for conspecifics density around occupied burrows. Population density was omitted in the analysis because it was highly correlated with masting events (Pearson's $r = 0.91$). Finally, all microhabitat environmental variables were also included, except for slope, because this measure was unavailable for most burrows.

For all models, all environmental variables were scaled except for factors. Prior to their inclusion in models, multicollinearity between all fixed effects was assessed (all variance inflator factors <3). Linearity of the relationship between independent variables and the logit of the response variable was evaluated by visually inspecting scatter plots between each predictor and the logit values. The dispersion, outliers, homogeneity, and non-independence of residuals were also checked for validity using graphical visualizations. All analyses were performed using R statistical software, version 3.6.3 (*R Core Team, 2020*).

## RESULTS

We located 219 different burrows on our sites (Fig. S2). Collars were fitted on 283 different individuals or 79% of all known site residents each year (Fig. S3), some of which received one on more than one year, for a total of 493 burrow-chipmunk associations, including 73 made from behavioral observations. Each year, we discovered "new" unidentified burrows, but as the study progressed, most collar-fitted individuals were tracked to a previously located burrow. Overall, 122 (55%) of burrows were occupied on more than one year, 92 of

**Table 1 Conditional logistic regression investigating burrow-site selection by eastern chipmunks, based on environmental variables within a 10 m radius plot around paired burrows and control points.**

| Effect | Estimate | SE | \|z\| | P |
|---|---|---|---|---|
| Canopy cover (%) | 0.057 | 0.045 | 1.28 | 0.20 |
| Herbaceous plants and shrubs (<1 m) cover (%) | 0.010 | 0.021 | 0.48 | 0.63 |
| Rocks cover (%) | −0.086 | 0.058 | 1.49 | 0.14 |
| Woody debris cover (%) | −0.068 | 0.057 | 1.20 | 0.23 |
| Number of logs (>2 m) | 0.258 | 0.137 | 1.89 | 0.059 |
| Number of refuges | 0.062 | 0.083 | 0.74 | 0.46 |
| **Slope (°)** | **0.256** | **0.073** | **3.51** | **<0.001** |
| Horizontal openness | 0.396 | 0.350 | 1.13 | 0.26 |
| Canopy height 1 | 4.771 | 4.891 | 0.98 | 0.33 |
| Canopy height 2 | 4.171 | 4.961 | 0.84 | 0.40 |
| Canopy height 3 | 3.080 | 4.676 | 0.66 | 0.51 |
| Canopy height 4 | 3.296 | 4.915 | 0.67 | 0.50 |
| Small hardwood tree density | −0.019 | 0.013 | 1.50 | 0.13 |
| Average DBH of large seed-producing trees | 0.166 | 0.117 | 1.42 | 0.16 |
| Large beech tree density | −0.104 | 0.109 | 0.95 | 0.34 |
| Large sugar maple tree density | −0.101 | 0.091 | 1.11 | 0.27 |
| Large red maple tree density | 0.119 | 0.257 | 0.46 | 0.64 |
| Yellow trout lily cover (%) | −0.047 | 0.049 | 0.95 | 0.34 |
| Carolina spring beauty cover (%) | −0.296 | 0.201 | 1.47 | 0.14 |

Notes:
The table presents all variables included in the full model, with their beta-coefficient, SE, z and P values. Canopy height 0 = low (reference level). Canopy height 1 = Mix of low and medium. Canopy height 2 = Medium. Canopy height 3 = Mix of medium and high. Canopy height 4 = High. Small trees have a diameter at breast height (DBH) ≤10 cm, while large trees have a DBH >31 cm. Seed-producing trees include American beech (*Fagus grandifolia*), red maple (*Acer rubrum*) and sugar maple (*A. saccharum*).
Analysis included 102 plots that were identified as a burrow-site at least once since 2012 and 102 control plots. Significant variables ($\alpha = 0.05$) are in bold.

which by at least two individuals, and 86 burrows (39%) were occupied by the same individual for at least two consecutive years.

The probability of a microhabitat being used as a burrow location increased with the slope of the terrain and with the number of logs it contained (Table 1), with a 29% increase in the probability of use with each additional degree of inclination, and 30% with each additional log, while holding all other predictors constant (Fig. 1).

During the 8-year study period, frequency of burrow occupancy ranged from 1 to 7 years (mean = 2.23, median = 2). The observed occupancy frequency distribution differed significantly from a random occupancy (tested using a zero-truncated Poisson distribution, $\chi_5^2 = 37.56$, $P = 0.007$, two-sided; Fig. S4). Burrows occupied for 1 year ($N = 98$) occurred more often than expected, whereas burrows occupied for 2 to 3 years ($N = 78$) occurred less often than expected, implying a non-random distribution.

Throughout all years ($N = 190$ burrows, 281 individuals), proportion of adult occupants in a burrow was positively associated with occupancy rate (coefficient ± SE = 1.35 ± 0.21, $z = 6.53$, $P < 0.001$), while proportion of female occupants was negatively associated with

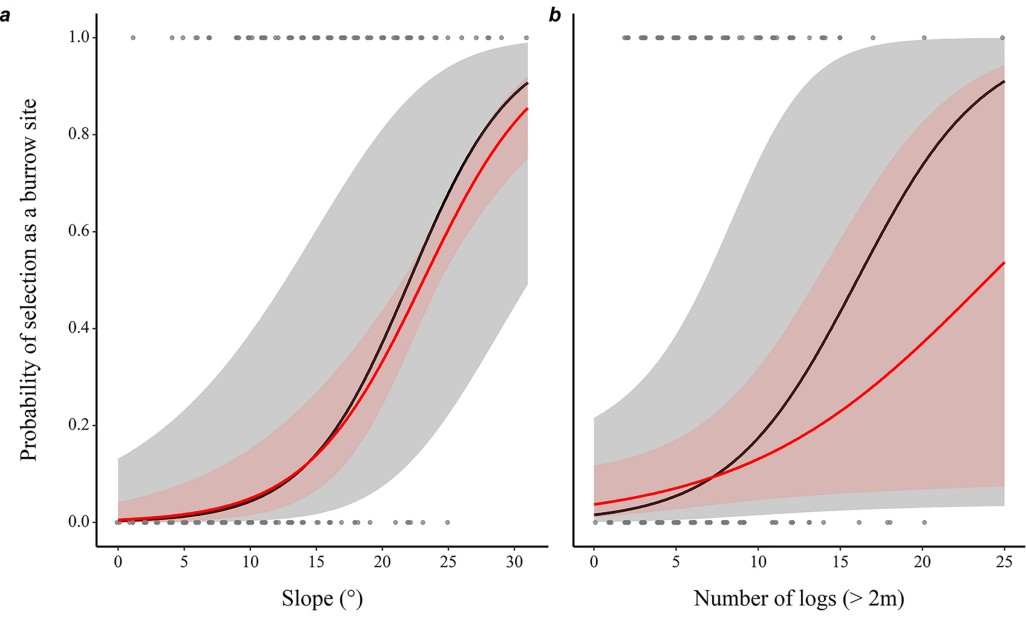

**Figure 1 Prediction of the likelihood of eastern chipmunks using a microhabitat plot as a burrow site.** (A) Effect of the slope (°) of the terrain. (B) Effect of the number of logs (>2m long) on the plot. Predictions were obtained from a conditional logistic regression including 102 burrow sites and 102 control plots, while holding all other predictor variables at their mean or modal (for categorical predictors) value. Black lines show predictions from the full model, while red lines show predictions from the model including only slope and number of logs. 95% confidence intervals are shown in each case. See Table 1 for statistical significance of other variables included in the full model.

occupancy rate ($-0.40 \pm 0.18$, $z = 2.25$, $P = 0.03$; Fig. 2). Burrow occupancy rate also decreased with microhabitat horizontal openness ($-0.27 \pm 0.08$, $z = 2.77$, $P = 0.006$) and shrub stratum cover ($-0.01 \pm 0.005$, $z = 2.24$, $P = 0.025$; Table S1).

Microhabitats of burrows mostly occupied by females were characterized by a taller canopy ($0.82 \pm 0.31$, $z = 2.66$, $P = 0.008$), less refuges ($-0.074 \pm 0.04$, $z = 1.83$, $P = 0.068$) and lower percentage ground cover of yellow trout lilies ($-0.04 \pm 0.01$, $z = 2.47$, $P = 0.014$) compared to microhabitats of burrows mostly occupied by males (Fig. 3 and Table S2).

Our analysis of 217 consecutive occupancy events showed that fidelity was significantly higher in non-mast than in mast years ($1.81 \pm 0.49$, $z = 3.68$, $P < 0.001$) (Table S3, Fig. 4 and Fig. S5). Fidelity also increased with the burrow occupant's age class ($0.49 \pm 0.25$, $z = 1.97$, $P = 0.049$), its average body mass ($0.06 \pm 0.03$, $z = 1.82$, $P = 0.068$), canopy cover above the microhabitat ($0.07 \pm 0.03$, $z = 2.66$, $P = 0.008$) and density of large red maples ($0.81 \pm 0.35$, $z = 2.30$, $P = 0.021$).

# DISCUSSION

## Microhabitat selection

We assessed populational and individual determinants of habitat selection in eastern chipmunks, in a system characterized by important inter-annual variations in food availability generated by masting deciduous trees. We found evidence for selection towards stable microhabitat characteristics, apparently favouring features which enhance either

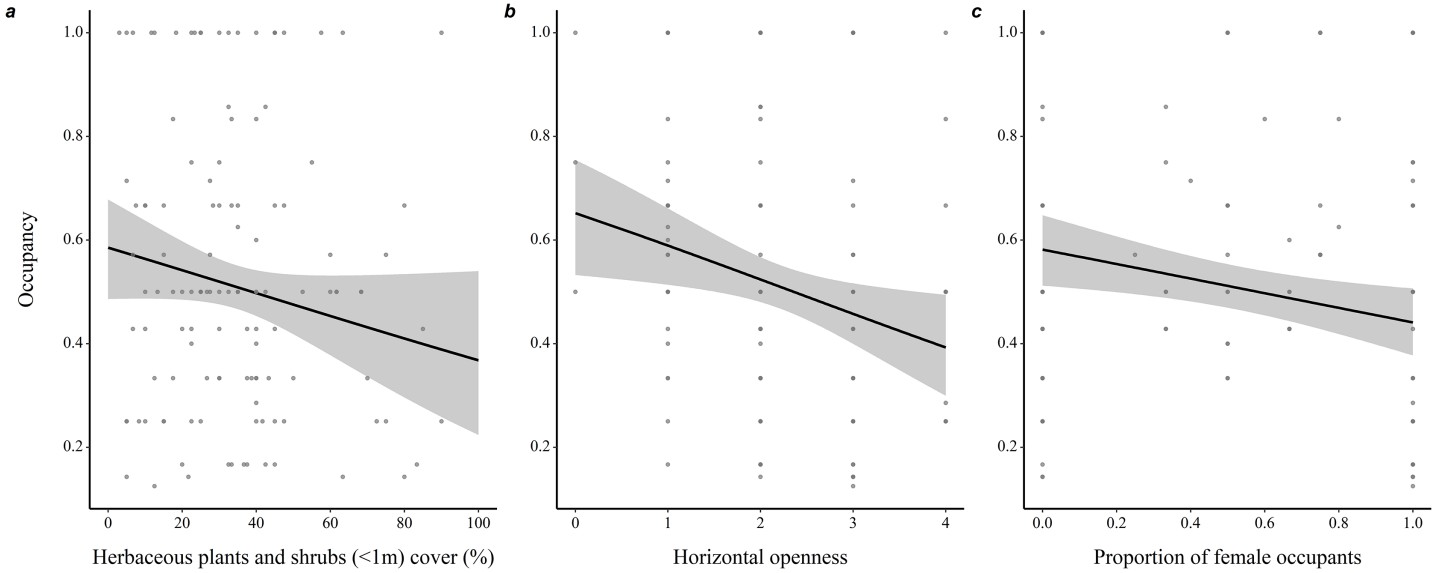

**Figure 2 Determinants of burrow-site occupancy rate by adult eastern chipmunks.** (A) Effect of the percentage cover of herbaceous plants and shrubs (<1m) in the microhabitat on occupancy rate. (B) Effect of the microhabitat's horizontal openness on occupancy rate. (C) Relation between the proportion of female occupants and occupancy rate. 95% confidence intervals are shown. Predictions were obtained from a logistic regression including 127 burrows occupied by adults only over the study period (2012–2019). When forming predictions, all other predictor variables were set at their mean or modal (for categorical predictors) value. Microhabitat is defined as a 10 m-radius plot surrounding a burrow.

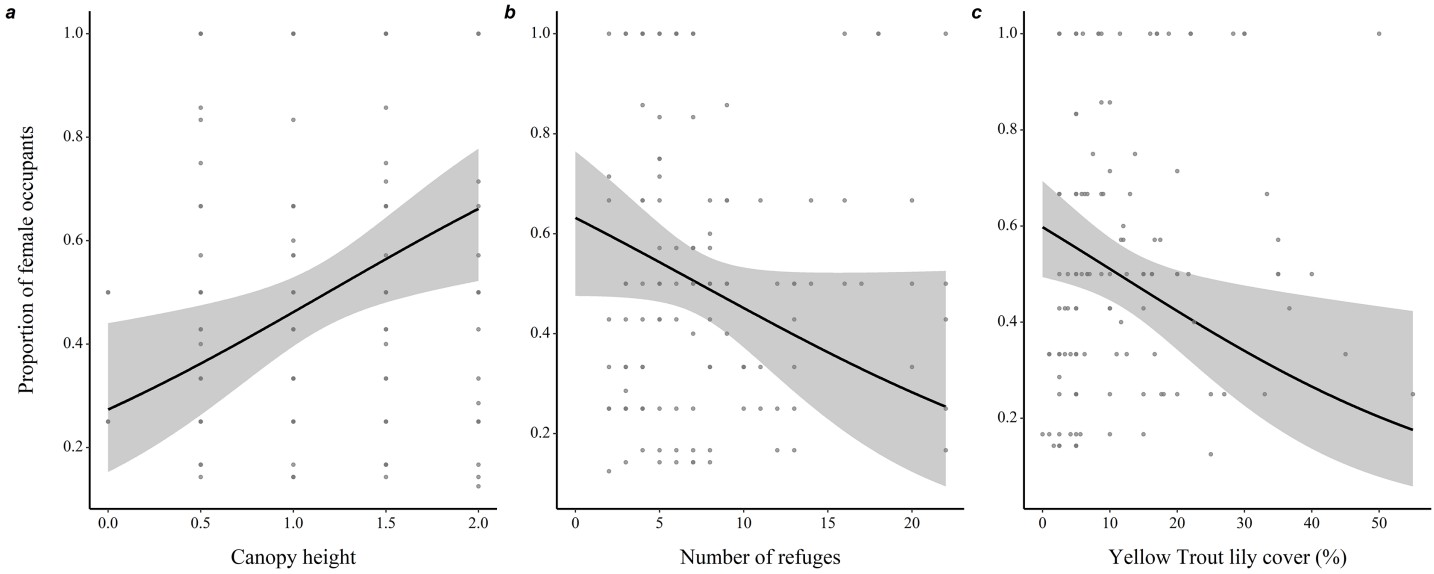

**Figure 3 Variations in burrow-site microhabitat features related to the sex-ratio of the burrow occupants in adult eastern chipmunks.** (A) Effect of the microhabitat's canopy height score. (B) Effect of the number of refuges in the microhabitat. (C) Effect of the percentage ground cover of yellow trout lily (*Erythronium americanum*) in the microhabitat. 95% confidence intervals are shown. Predictions were obtained from a logistic regression including 127 burrows occupied by adults only over the study period. When forming predictions, all other predictor variables were set at their mean or modal (for categorical predictors) value. Microhabitat is defined as a 10 m-radius plot surrounding a burrow.

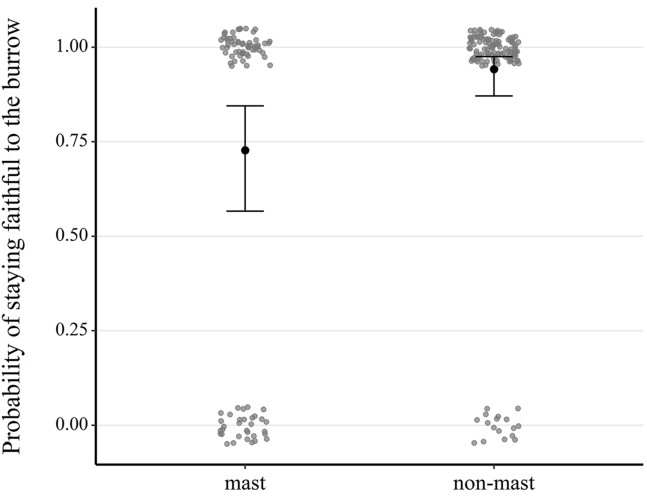

**Figure 4 Comparison of the probability of next-year fidelity to a burrow during mast and non-mast years in eastern chipmunks.** Analysis included 217 occupancy events for which the resident was alive the following year and the location of their burrow reassessed. The model's predictions are illustrated by black dots, and 95% confidence intervals are represented by brackets. When forming the prediction, all other predictor variables were set at their mean or modal (for categorical predictors) value. Raw data are shown as scattered dots for each case possibility (mast-faithful, mast-unfaithful, non-mast-faithful, non-mast-unfaithful) to allow visualization of their distribution.

burrow construction, thermal conditions, or predator avoidance over food availability. First, burrow microhabitats were located on steeper slopes than expected based on availability, a trend seen in the dens and burrows of other mammals, including ground squirrels (*Spermophilus* sp.; *Barker & Derocher, 2010*), yellow-bellied marmots (*Marmota flaviventris*; *Bednekoff & Blumstein, 2009*), arctic foxes (*Alopex lagopus*; *Szor, Berteaux & Gauthier, 2008*) and black bears (*Ursus americanus*; *Reynolds-Hogland et al., 2007*). Previous studies on eastern chipmunks also confirm this tendency (*Svendsen & Yahner, 1979*; *Mahan & Yahner, 1996*). Slanted terrain likely provides advantageous soil conditions for burrow construction because of better drainage and earlier snow melt, depending on exposition (*Svendsen, 1976*; *Karels & Boonstra, 1999*). Alternatively, a burrow located in a steeper terrain could offer better isolation from the cold during hibernation by sinking further from the ground surface, considering that the frost front's depth in winter approximates or exceeds the average depth of chipmunk burrows (*Panuska & Wade, 1956*; *Landry-Cuerrier et al., 2008*). Finally, slope could also be related to predator avoidance, as it may enhance visibility range and improve vigilance (*Barker & Derocher, 2010*), especially when combined with a low density of the shrub stratum.

Second, we showed that burrow microhabitat contained a high quantity of coarse woody debris, a feature particularly associated with small mammals' presence, abundance and diversity (*e.g.*, *McComb, 2009*; *Fauteux et al., 2013*; *Sullivan & Sullivan, 2019*), as they play diverse functional roles in predation avoidance. For instance, woody debris facilitates predator detection by prey by allowing the latter to rise above dense vegetative ground

cover (*Machutchon & Harestad, 1990*; *Mahan & Yahner, 1996*), and reduces detection of prey by predators, because movement over woody debris is faster and minimizes the noise produced compared to movement in dry leaf litter (*Roche, Schulte-Hostedde & Brooks, 1999*; *McCay, 2000*). Furthermore, woody debris increases the probability of escaping a predator attack by acting as territorial landmarks for orientation, by providing refuges and by increasing microhabitat structural complexity (*Barry & Francq, 1980*; *Waldien, Hayes & Huso, 2006*). Chipmunks are known to spend more than 25% of their daily time-activity budget perched on woody debris, in vigilance, eating, grooming or vocalizing (*Landry-Cuerrier, 2008*) and are frequently observed moving along branches and logs (*Waldien, Hayes & Huso, 2006*), particularly when moving in open canopy areas (*Zollner & Crane, 2003*), consistent with the hypothesis that woody debris plays a role in protection from predators. Overall, chipmunks inhabiting burrows surrounded by abundant woody debris likely enjoy a safer, more complex microhabitat with many refuges and perch sites, as well as reduced vigilance requirements and thus more profitable foraging under low predation risks (*Lima & Dill, 1990*; *Landry-Cuerrier, 2008*).

Furthermore, our results show that burrows with lower horizontal openness and a lower percentage of ground cover by herbaceous plants and shrubs had higher occupancy rates. These findings are in line with previous studies that showed clear selection for open understory and closed overstory habitats in chipmunks (*Svendsen & Yahner, 1979*; *Bowers, 1995*). Dense under story cover, especially herbaceous and shrub cover, restrict horizontal visibility near the ground, therefore impeding predator detection (*Svendsen & Yahner, 1979*). While vegetative cover has been shown to provide a hiding place and visual concealment from predators, many small mammals conversely avoid under story growth that obstruct their surroundings' view (*Bakker, 2006*; *Bednekoff & Blumstein, 2009*; *Guo et al., 2020*). Besides shorter flight initiation distance due to delayed visual detection of attacking predators, dense ground cover can reduce escape speed (*Schooley, Sharpe & Van Horne, 1996*; *Guo et al., 2020*) and create visual and auditory cues for predators (*Bakker, 2006*). Since all ground cover is probably both obstructive and protective to some extent, there may be a trade-off in its use by prey (*Lazarus & Symonds, 1992*; *Sharpe & van Horne, 1998*).

It has been suggested that the benefits associated with anti-predator structural features in a microhabitat outweigh those of localized food access (*Landry-Cuerrier, 2008*). This is consistent with our findings that food availability in microhabitat did not influence burrow-site selection. Mast-seeding is possibly too spatiotemporally variable; thus, it is likely impossible for chipmunks to select a burrow location that would always maximize access to all types of food resources. The optimal strategy would therefore be to select a microhabitat according to more temporally stable and spatially heterogenous criteria, such as predation risk avoidance, and travel at a larger scale to reach rich food patches when necessary (*Yahner, 1978a*).

## Microhabitat occupancy

We found that chipmunks occupied burrow microhabitats in a non-random pattern and that the age and sex of the occupants was related to occupancy rate, which could point to

differential competitive abilities in obtaining a higher quality microhabitat. Burrows with lower occupancy rates were proportionally more occupied by juveniles than adults, which was expected given both their inexperience in identifying appropriate habitat features and competitive exclusion from established adult residents (*Yahner, 1978a*; *Couchoux et al., 2021*). While adult chipmunks systematically chase and exhibit aggressive behavior towards any conspecific that trespasses into their core area (*Dunford, 1970*; *Getty, 1981*), juveniles never display territoriality and even tolerate intruders (*Elliott, 1978*; *Yahner, 1978b*). Furthermore, juveniles' search for their first burrow upon natal dispersal inevitably takes place in a context of high population density and reduced burrow availability (*Bergeron et al., 2011*), making it likely that establishment occurs at any vacant burrow, even if the microhabitat is not optimal (*Panuska & Wade, 1956*; *Elliott, 1978*). Therefore, low-quality sites may be used when population density is high by young and naive individuals, until they can shift to a higher quality microhabitat vacated by an adult resident (*Yahner, 1978a*).

While it could be expected that juveniles would be prevalent in burrows with lower occupancy rates (*i.e.*, presumably low-quality burrows), this trend was also observed for females, which could partly be explained by dominance and/or competitive exclusion from higher-quality microhabitats by males. Dominant behavior in chipmunks should be spatially variable, with a reversal of an individual's dominant or subordinate status depending on the proximity of its burrow (*Dunford, 1970*; *Couchoux et al., 2021*). However, males may be more frequently involved in agonistic encounters, as dominance hierarchies can be established between males participating in mating bouts throughout the breeding season (*Elliott, 1978*). Alternatively, occasional territorial bequeathal of the natal burrow to one offspring may explain the greater presence of females in less-occupied burrows. Such a behavior was observed in 30% of breeding females in North American red squirrels (*Tamiasciurus hudsonicus*; *Berteaux & Boutin, 2000*), a species ecologically close to eastern chipmunks. This behavior is hypothesized to be a form of parental investment, as foregoing the hazards of dispersal increases juvenile survival chances (*Berteaux & Boutin, 2000*). If this phenomenon is present in chipmunks, it could lead to the observed prevalence of females in less-occupied burrows. Intersexual differences in microhabitat use can also reflect the asymmetry in reproductive roles typically found in small mammals. Females could be constrained to microhabitats containing features more suitable to young rearing, while male distribution is only restricted by access to breeding partners (*Emlen & Oring, 1977*; *Hwang, Larivière & Messier, 2007*). In that sense, intersexual microhabitat partitioning may be adaptive by reducing intraspecific overlap in resource use (*Morris, 1984*).

## Microhabitat fidelity

Year-to-year burrow-site fidelity has been described as extremely high in adult chipmunks (*DeCoursey & Krulas, 1998*; *Marmet, Pisanu & Chapuis, 2009*), although some authors noticed shifts between burrows (*Elliott, 1978*; *Yahner, 1978a*). We report a site fidelity rate of 76% across 7 years and 242 instances of reassessing a burrow's occupant identity on two consecutive years. Fine acquaintance with the unique specificities of a habitat and its social

environment, developed through long-term occupancy, is considered to procure major benefits, such as knowledge of foraging routes and food location, effective escape from predators, territorial dominance and reduced agonistic interactions with neighbors (*Piper, 2011*; *Acker et al., 2017*). For instance, *Clarke et al. (1993)* found that individuals fleeing within their home range took half the time and covered half the distance before reaching a refuge compared to individuals fleeing outside their home range, indicating there are familiarity advantages to remain at the same burrow. The positive associations we found between an individual's age and the probability of its faithfulness to a burrow is in line with these familiarity advantages gained through prolonged site-fidelity.

Fidelity also increased during non-mast years, which contradicted our initial prediction that chipmunks would tend to remain in their burrow following a massive peak in seed availability to benefit from the hoarded food. It would thus be intuitive to assume that the value of a burrow's larder hoard, and its owner's incentive to maintain possession, increases with its size. This does not seem to be the case: an individual associated to a given burrow in June on a given year is more likely to use the same burrow the following year when no mast occurred during the autumn, compared to when a mast did occur. *Humphries et al. (2002)* showed that chipmunks can accumulate an entire winter's worth of energy within a few days of intensive mast crops hoarding, but that they cease aboveground activity and initiate hibernation even when additional food storing is still possible. This suggests that the relation between hoard size and energy gains may not be linear. Prolonged storage time could also be detrimental: stockpiled seeds in chipmunk burrows are vulnerable to germination, rotting and insect infestation, which can render the whole hoard worthless (*Elliott, 1978*).

An explanation for the higher frequency of burrow changes in mast years may be that individuals prefer to disperse in a low-risk context; the predictable abundance of food at a spatial scale much larger than their dispersal ability might mitigate the costs of searching for and settling in a new microhabitat. In contrast, high site fidelity rates on non-mast years (>87%) are likely explained by the high population density during these years. Two concomitant reproductive episodes normally occur in the 8 months preceding a non-mast active season, causing a drastic increase in population size (*Bergeron et al., 2011*) and thus presumably low burrow vacancies. Site fidelity therefore appears to be the result of low availability of sites rather than an adaptive habitat selection decision (*Martinez et al., 2017*). Indeed, our data indicates that annual trap diversity (*i.e.*, the number of different traps where a given individual is captured), a measure for exploration behavior in our study system (*Paquette et al., 2020*), was significantly lower on non-mast years, namely at high density (Fig. S6). It is therefore possible that chipmunks reduce the range of their movements and thus their search of potential vacant microhabitats when population density is high. These observations support the premise that attaining incumbency at a burrow occurs on a stochastic and opportunistic basis, not a dominance-related one.

## CONCLUSIONS

Precise description of patterns of habitat selection is essential to understanding a species' ecology. Here we found that chipmunks show some selective behavior for specific

microhabitat features related to burrow placement. Furthermore, interindividual differences in sex and age may lead to differential selection of certain microhabitat features and/or differential access to microhabitat quality. In a context of high density during non-mast years, individuals that might otherwise prefer to switch burrows seem constrained and unable to do so. A decrease in density during mast years due to mortality, emigration, and a null natality, would allow them to express more choice options, using cues such as beneficial microhabitat features, and/or their personal perceived value of the space, possibly assessed through site familiarity advantages, to decide whether and where to resettle. Because habitat selection impacts all aspects of an individual's life, our results can serve as a basis for further investigation of several other questions of interest. Examining the consequences of microhabitat selection on the reproductive success or survival of the resident is a logical next step. Overall, this work adds to the fundamental knowledge accumulated on a model species of small mammals and emphasizes the need to consider variation in environmental context and individual characteristics in habitat selection studies.

## ACKNOWLEDGEMENTS

We thank Nature Conservancy of Canada and Appalachian Corridor for allowing this long-term project to be conducted on their protected areas. We also thank two anonymous reviewers for helpful comments on the manuscript. We are grateful to the many graduate students, field and lab assistants who helped with data collection and analyses since 2005.

### Funding

This research was funded by Natural Sciences and Engineering Research Council of Canada (NSERC) Discovery grants to Dany Garant, Patrick Bergeron and Denis Réale and by a team research grant from the Fonds de Recherche du Québec—Nature et technologies (FRQNT) to Dany Garant, Patrick Bergeron and Denis Réale. Camille Gaudreau-Rousseau was supported by a FRQNT Master's Research Scholarship and a scholarship from Université de Sherbrooke. There was no additional external funding received for this study. The funders had no role in study design, data collection and analysis, decision to publish, or preparation of the manuscript.

### Grant Disclosures

The following grant information was disclosed by the authors:
Natural Sciences and Engineering Research Council of Canada.
FRQNT Master's Research Scholarship.
Université de Sherbrooke.

### Competing Interests

Dany Garant and Patrick Bergeron are Academic Editors at PeerJ. The other authors declare that they have no competing interests.

## Author Contributions

- Camille Gaudreau-Rousseau conceived and designed the experiments, performed the experiments, analyzed the data, prepared figures and/or tables, authored or reviewed drafts of the article, and approved the final draft.
- Patrick Bergeron conceived and designed the experiments, authored or reviewed drafts of the article, provided logistic support in the field, and approved the final draft.
- Denis Réale conceived and designed the experiments, authored or reviewed drafts of the article, provided logistic support in the field, and approved the final draft.
- Dany Garant conceived and designed the experiments, authored or reviewed drafts of the article, provided logistic support in the field, and approved the final draft.

## Animal Ethics

The following information was supplied relating to ethical approvals (*i.e.*, approving body and any reference numbers):

Animals were captured and handled in compliance with the Canadian Council on Animal Care, under the approval of the Université de Sherbrooke Animal Ethics Committee.

## Field Study Permissions

The following information was supplied relating to field study approvals (*i.e.*, approving body and any reference numbers):

Field experiments were approved by the Ministère de la Forêt, de la Faune et des Parcs du Québec–Direction de la gestion de la faune de l'Estrie, de Montréal, de la Montérégie et de Laval pour la capture des animaux sauvages à des fins scientifiques, éducatives ou de gestion de la faune (SEG) (#2017-05-01-102-05-S-F, #2018-04-20-103-05-S-F, #2019-04-30-104-05-S-F).

## Data Availability

The raw data and code are available in the Supplemental Files.

## Supplemental Information

Supplemental information for this article can be found online at http://dx.doi.org/10.7717/peerj.15110#supplemental-information.

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
