# Peer review of "Environmental and individual determinants of burrow-site microhabitat selection, occupancy, and fidelity in eastern chipmunks living in a pulsed-resource ecosystem"

_PeerJ, doi:10.7717/peerj.15110_

## Round 0.1 · original submission · Major Revisions

Dear authors,

The two reviewers enjoyed your manuscript. So did I. It is well written, the experimental design is solid, and the results are interesting. However, they also raised a number of major comments that need to be carefully addressed before further consideration for publication (i.e. manuscript would gain in appeal if significantly shortened, code should be proof reread, alternative statistical analyses could be considered). See also the detailed report. For these reasons, I advise major revisions.

Mathieu Lihoreau

(Editor)

Reviewer 1 ·

Basic reporting

I believe the authors have done an excellent job with this study. In fact, the authors may have been too ambitious in their writing! The text is ~10000 words and while PeerJ seemingly does not have a word limit, I would argue that if authors cannot get their stories across in ~7000 words or less (the word limit in most venues) there is potentially an issue (plus it is hard for the reader to stay engaged in very long articles). I got the sense that the authors in this case are very well versed in the system and the literature review is thorough, however, there were places where the authors seemingly added too much information and could condense down to make the paper shorter. I have provided an attached document with suggested changes to help the authors cut down words and improve clarity and flow. These edits did not take part in my overall decision on the quality of the science contained herein!

For the tables please make sure the bold text goes all the way across for the significant items for ease on the reader. Also please mention site (as seen on Table S1) in your habitat selection methods because this was not described in the modelling section.

Experimental design

no comment - the authors did excellent in this regard

Validity of the findings

I would recommend the authors simplify the code to only the things they have done for the manuscript. For example, you have both conditional logistic regression and logistic regression in the code but only use one in the manuscript. Similarly, you could condense the data cleaning and exploring steps and files to what is essential for replication.

Annotated reviews are not available for download in order to protect the identity of reviewers who chose to remain anonymous.

Reviewer 2 ·

Basic reporting

I found the paper to be well written, with sufficient literature and background information to motivate the study and to support the authors' a priori hypotheses. The authors provide all data and code used to analyze the data as supplementary files.

Experimental design

The authors address important and meaningful questions related to how selection of chipmunk burrows varies among individuals and with population density and environmental variability (mast/non-mast years). Although I have a few suggestions for the authors to consider, the analytical methods they use seem largely appropriate and they are generally well described.

**Data and code:**
I really appreciate that the authors provided all of their data and code. I didn't look through the code too closely, but I did try to run several of the R scripts. I *think* there are a few typos in the scripts that prevent them from running. Specifically, the following line needs to be updated:

# From here::i_am(file.path("Script corvif.R")) # To the line below:
here::i_am(file.path("Script corvif.v2.R"))

Same goes for the following line of code in 2-variables_choices_obj1.Rmd
# Change: source(here("Visualisation covariation.R")) # to
source(here("Visualise.covariation.v2.R"))

Even after making these changes, I was unable to "knit" some of the scripts (though I was able to step through the different code chunks successfully). So, some more work may be needed to make sure all of the scripts can be run successfully by readers. Otherwise, the code is fairly well documented. I also appreciated the efforts made by the authors to explore the potential for collinearity among their predictors and also their attempts to evaluate model goodness of fit.

Below, I offer a few thoughts on the statistical methods used by the authors to analyze their data.

**Preliminary analyses**:
Looking through the authors' scripts, it is clear that they spent quite a bit of time exploring issues related to collinearity among their potential predictor variables. Again, I really appreciate the thoughtfulness here. On the other hand, they also appear to have used several simple t-tests to help guide the selection of appropriate predictor variables. They also use stepwise selection to choose final models. In doing so, the authors risk overfitting their data. The p-values for the variables that remain will be biased low, and the chosen model will almost surely preform worse if applied to new data. Essentially, the data are used twice (once to choose potential predictors via the t-tests and stepwise selection algorithms and then again to test hypotheses in their multivariate regressions models). These issues are well documented in both the statistics and ecological literature - see e.g., Whittingham et al. (2006), Fieberg and Johnson (2015) and in Frank Harrell's regression modeling strategies book.

Fieberg, J., & Johnson, D. H. (2015). MMI: Multimodel inference or models with management implications? The Journal of Wildlife Management, 79(5), 708–718.

Harrell Jr, F. E. (2015). Regression modeling strategies: With applications to linear models, logistic and ordinal regression, and survival analysis. Springer.

Whittingham, M. J., Stephens, P. A., Bradbury, R. B., & Freckleton, R. P. (2006). Why do we still use stepwise modelling in ecology and behaviour?. Journal of animal ecology, 75(5), 1182-1189.

The authors have a rich data set, and the results are interesting and should definitely be published. Yet, there is a downside to the large number of hypothesis tests and models that were fit - I would tend to use those approaches more for "hypothesis generation" than hypothesis testing (it also leads to really long R scripts that are somewhat difficult to follow).

I would have much preferred to have seen the authors fit a single model with all covariates of interest & then use the estimates and their standard errors to make conclusions about statistical and biological significance.

*Microhabitat selection*
The authors fit conditional logistic regression models to test hypotheses regarding burrow site selection, which is pretty standard for the type of data they collected. They also fit "non-conditional" logistic regression models to evaluate site-specific effects, which they report in supplemental material. This latter analysis doesn't make sense to me (even though I am familiar with how conditional logistic regression models can be fit using standard software for unconditional logistic regression by taking the difference between covariate values for the matched observations).

In paired or matched data, it is not possible to include predictors that do not vary within a stratum as the predictor variable will be constant across all possible choices in the choice set. Instead, what is usually done, and what I would recommend (if site-specific effects are of interest), would be to consider interactions between site-level covariates and other covariates in the model that do vary within the matched set. In this way, one can ask questions about whether site-level covariates modify the effect of other variables. As one example, the authors might have a look at Forester et al. 2009 (figure 4). They included interactions between four time of day harmonics (sin/cos terms) and landcover variables to evaluate how selection for these landcover types varied with time of day.

Forester, J. D., Im, H. K., & Rathouz, P. J. (2009). Accounting for animal movement in estimation of resource selection functions: sampling and data analysis. Ecology, 90(12), 3554-3565.

One could also interact the sex of the inhabitant with environmental covariates to determine if males/females' select for different characteristics.

*Microhabitat occupancy*
The authors state that they use a "proportional logistic model" (I'm not sure what this is & a google search was unhelpful) to analyze what are essentially binomial (or a set of Bernoulli) observations (X = number of years occupied; N = number of years available to be occupied = number of years since the burrow was first detected). It is not clear to me why they did not just use logistic regression with response = 1 if occupied and 0 if not (for each year the burrow was known), which would allow for time-varying predictors (e.g., environmental variables that changed from year-to-year). If the authors were concerned about non-independence, they could consider adding random intercepts for "burrow" and potentially, random slopes for variables that vary over time at the same burrow. Rather than include proportion of adult occupants and proportion of female occupants, one could include an estimate of density of adults and females in the study site.

It might even be possible to use a single analysis to look at microhabitat selection by including "available" (matched sites) as unoccupied locations rather than conducting 2 separate analyses (one looking at burrow site-selection and one looking at burrow occupancy). Yet, I can also see value in conditioning the analysis on sites that were used as burrows at some point in time.

*Proportional binomial model with sex ratio (females/males)* (lines 354-361)
Again, I am not sure what a proportional binomial model is, but I don't see why the authors could not fit a model with sex as the response variable (say = 1 if the burrow was occupied by a male and 0 if occupied by a female): glm(male ~ microhabitat variables, family = binomial()).

Validity of the findings

The authors have gathered a rich data set capturing multiple mast and non-mast years over a 7 year time span. The conclusions seem reasonable in light of the data and are linked to the authors original research questions.

Additional comments

Specific comments:
1. Lines 25-26: I would delete “non-random” and change “scale” to scales (plural)
2. Line 103: “Longevity of chipmunks varies between 1.5 and 3 years, but ranges up to 8 years” – I found this statement to be confusing. How were these numbers arrived at? Does the 1.5 – 3 represent a confidence interval for the mean and the 8 represent a maximum value? More detail here would be useful.
3. Line 201: individuals were considered residents only if they were captured more than 5 times. That seems like a pretty extreme criterion. How prevalent were non-residents? Also, have the authors looked to see if capture and recapture influences subsequent behaviors?
4. Line 206: individuals had to be observed carrying food into the same burrow at least 3 times in the same week to assign a burrow? That also seems extreme. Do chipmunks cache food in multiple burrows?
5. Line 307: I assume there is a typo here – shouldn’t this be: “Population density WAS omitted”?
6. Line 310-311: how was the significance of the random effect evaluated? More often than not, I would tend to include the random effect even if there is not enough evidence to conclude that the variance of the random effect is different from 0. See e.g., discussion regarding testing significance of random effects here: GLMM FAQ (bbolker.github.io)
7. Line 314, lines 319-321: How was linearity evaluated? How were other assumptions evaluated? Standard residuals for logistic regression models should not have constant variance (though Pearson residuals should have constant variance). More information would be helpful here. The authors might have a look at the check_model() function in the performance package in R.
8. Lines 331-333: Technically, this sentence should also state "while holding all other predictor variables constant". Also, I assume Figure 1 was formed by holding all other predictors at their mean or modal values. If so, that should be stated.
9. Lines 333-334: the methods section and table legend suggest that stepwise backwards selection was used to eliminate non-significant variables - but, the table includes many variables that are not statistically significant at an alpha of 0.05. I find that confusing and am not sure how to reconcile the table with the description. And, my strong preference would be to just report results from the full model (rather than apply backward stepwise selection).
10. Line 335: suggest "due to the effect of slope" (drop "the")
11. Lines 350-353: since the sample is largely dominated by adults, it make sense that an adults-only model would look just like the global model. This paragraph could probably be deleted.
12. Line 374: "Burrows occupancy rate was related to the age and sex of their occupants" - this statement doesn't make any sense to me. If a burrow is occupied by a male/female or a young/old individual, it is occupied. So, how can occupancy rate depend on the characteristics of the individuals that occupy the den?
13. Line 401, 454: suggest "prey" rather than "preys"
14. Table 1: again, it is not clear how to reconcile the large number of non-statistically significant results with the fact that the model was chosen by backwards selection.
15. Figures 1 and 2: the legends should provide information about the non-focal predictors in the model - I assume they were set to mean or modal (categorical predictors) values when forming the predictions.
16. Figure 2 legend: what is a "proportional logistic regression", and can you provide a reference?
17. Figure 3: "Variations in burrow-site microhabitat features according to the sex-ratio of the burrow occupants in adult eastern chipmunks." I find this difficult to understand. It equates features of a burrow with a sex-ratio.
18. I would change this to a bar graph. The points (0's and 1's) do not tell us anything beyond the proportion and sample size in each group.

---

## Round 0.2 · Minor Revisions

I have been asked, as one of the PeerJ Section Editors (Ecology) to take over the handling of this manuscript.

This manuscript has had a very robust review process, and I very much appreciate the detailed assessment from the reviewers and the equally detailed and helpful responses from the co-authors. At this time I see this as a minor revision.

In particular, I ask the co-authors to please note the following from Rev. 2:

"To do so, one could fit a logistic regression model to the non-aggregated data (occupancy yes/no in each year) with a time-varying predictor that captures whether the site was occupied the previous year interacted with sex or age class of the occupier. This would allow one to test whether a site that was used in the previous year was more likely to be used than a site that was vacant last year – and, whether the answer depends on the occupier in the current year. This again demonstrates the potential advantages of fitting logistic regression models to the yearly data rather than aggregating the data first."

Please consider this suggestion carefully in your revision and provide a rationale for your decision.

All other suggestions from both reviewers are comparatively minor, and I don't think the suggestion above rises to the level of a major revision. As such, I don't expect to send this out for another round of review, unless Rev. 2 contacts me ([email protected]) and would like to see it one final time.

Thanks again to all involved for your work on this. I look forward to seeing your revisions.

Reviewer 1 ·

Basic reporting

The authors have done an awesome job in their revision! The manuscript is much easier to read, there is still sufficient background/context and cited literature, and they have kept a great narrative to their work.

Experimental design

The experimental design description, investigation is rigorous, and research questions are well defined.

Validity of the findings

Findings are valid.

Additional comments

The authors have done an awesome job in their revision! I only have a few minor text based comments for the authors to consider:

Line 16 - 18: Suggest that the wording is changed to: "Our results indicate that chipmunks select microhabitats with a greater amount of woody debris and greater slopes. More frequently occupied burrows also had a lower shrub stratum density, were less horizontally opened and were predominated by males." This changes the word more to greater and places things in the past tense.

Line 291: Please spell out 86 - statements should not start with a number.

Line 296-298: You can likely drop this statement about the R-squared - the value is listed on your table caption and we can see that the estimate for slope is the driving force in the model therefore stating this is likely redundant.

Lines 306-319: The values in these paragraphs (z, P and estimates with confidence intervals) are also in the Tables therefore it seems like you are repeating results. I would suggest taking the values out and just citing the relevant table at the end of each statement.

Line 346:detection of prey - not "preys"

Line 348 - period before the citation?

Lines 362-367: I believe understory and overstory are single words with no hyphen

Line 405: (Tamiasciurus hudsonicus) (Berteaux & Boutin, 2000) can be written as (Tamiasciurus hudsonicus; Berteaux & Boutin, 2000). This removes the awkward double parentheses.

I want to thank the authors for their efforts on the revision and look forward to seeing this in print!

Reviewer 2 ·

Basic reporting

I plan to post most of my comments in the Additional Comments box and just highlight how they fit into these 3 sections.

The paper is well written. One minor concern here - see comments regarding "proportional logistic models".

Experimental design

See additional comments regarding the authors' approach to data analysis and suggestions for improving how the results are presented. I strongly recommend presenting coefficients from the full model rather than coefficients from a mix of models resulting from backwards stepwise selection.

Validity of the findings

No comment

Additional comments

I appreciate the authors efforts to improve clarity of the manuscript (following reviewer #1’s suggestions), the choice to drop one of the supplemental analyses, and efforts to improve the readability and reproducibility of their coded analyses. I continue to believe the paper has significant merit, though I do not fully agree with some of their responses to the initial reviews.

I feel they were a little too dismissive of the concerns related to the model selection and its impacts on the strength of inference that can be drawn from the data. In particular, using t-tests and other exploratory methods as well as backwards stepwise model selection to determine which predictors to include in a final model has long been known to be problematic when it comes to inference; p-values will be biased low, confidence intervals will be too narrow, etc for the predictors that show up and remain in the final model. Although these types of steps are common, the results should be viewed as more exploratory (i.e., used to generate rather than test specific hypotheses).

One way to reduce these concerns is to do less model selection, which is one reason why I suggested presenting the estimates and confidence intervals associated with the full model. In addition, I find the tables of coefficients to be particularly misleading since they report coefficients estimated from several different models all in the same table – and the interpretation of the coefficients changes whenever other predictors are added or dropped (see e.g., Cade 2015. Model averaging and muddled multimodel inferences. Ecology 96:2370-2382). I would strongly recommend reporting coefficients, confidence intervals, and p-values from the full model instead.

The authors continue to refer to their logistic regression models as “proportional logistic models”, which is confusing since, as their response made clear, these are just logistic regression models specified by giving the glm function the proportion of trials with successes and the total number of trials as weights. As the help function for glm details, the exact same model could be fit by sharing the number of successes and number of failures, or by using a series of Bernoulli trials (occupied or not in each year). The advantage of this last approach, which is what I suggested in my initial review, is that time-varying predictors could be included to capture annual variation in factors related to occupancy status. Without time-varying predictors, this approach should give identical coefficients, standard errors, etc as the approach the authors used. Thus, I would recommend dropping “proportional” and just refer to logistic regression models.

Specifying the logistic regression model using aggregated data (i.e., the authors approach using proportions and the number of trials as weights or, alternatively, using the number of successes and number of failures) does make it possible to include a predictor that considers the proportion of adults or proportion of females occupying the burrow over time – but I find these models to be really strange and the results to be extremely awkward to interpret. For example, take the statement that, “Burrows occupancy rate was related to the age and sex of their occupants” - how would one predict whether a burrow is likely to be occupied using the age or sex of the occupants? You can’t - this type of statement makes no sense to me. What the authors could do instead is evaluate whether males are more likely than females to use a site that was occupied last year. To do so, one could fit a logistic regression model to the non-aggregated data (occupancy yes/no in each year) with a time-varying predictor that captures whether the site was occupied the previous year interacted with sex or age class of the occupier. This would allow one to test whether a site that was used in the previous year was more likely to be used than a site that was vacant last year – and, whether the answer depends on the occupier in the current year. This again demonstrates the potential advantages of fitting logistic regression models to the yearly data rather than aggregating the data first. I would recommend the authors reach out to a statistician if this doesn’t make sense.

Other minor suggestions:

1. The authors dropped the description of the methods they used to evaluate the assumptions of their models based on feedback from reviewer #1 that “I don’t think you need to state this for the reader. Assumptions always need to be met for a model to be valid so it is intuitive that you had to test for them.” I strongly disagree. Yes, assumptions should always be evaluated, but that does not mean that this is standard practice. In addition, it is important to know *how* assumptions were evaluated.
2. Line 17-18: “More frequently occupied burrows have a lower shrub stratum density, are less horizontally opened and less occupied by juveniles and females.” This is another example of a really awkward statement – “More frequently occupied burrows are less occupied by juveniles and females.” What does it mean for a more frequently occupied burrow to be less occupied by juveniles and females?
3. Line 121: since this is a prediction, I would state this as “we predicted burrows would be located”
4. Line 124: suggest, “We also predicted that burrow microhabitats would contain”

---

## Round 0.3 · accepted · Accept

This article has been through a thorough review process, and the reviews have been helpful at all stages. The co-authors have done a good job of responding the the reviewers. The final stage of revisions were quite minor, with one somewhat substantial suggestion from Rev. 2 in particular. The co-authors have given a good response to that suggestion, and this final round of revisions have taken this manuscript to a point where it is real for publication.

Please note that I mentioned in my previous decision letter (minor revisions) that Reviewer 2 could contact me if they wanted to take one final look at the revisions. To my knowledge, Reviewer 2 has not contacted me, and thus I am making this decision based on what I have seen of the review process and the excellent responses from the co-authors.

Thank you to all involved for your excellent contribution to PeerJ and to the field.